# Analysing similarities between legal court documents using natural language processing approaches based on transformers

**Raphael Souza de Oliveira**[1,2], **Erick Giovani Sperandio Nascimento**[2,3]*

**1** TRT5 - Regional Labour Court of the 5th Region, Salvador, Bahia, Brazil, **2** Stricto Sensu Department, SENAI CIMATEC University, Salvador, Bahia, Brazil, **3** Surrey Institute for People-Centred Artificial Intelligence, Faculty of Engineering and Physical Sciences, University of Surrey, Guildford, United Kingdom

☯ These authors contributed equally to this work.

* erick.sperandio@surrey.ac.uk

**Data availability statement:** All data files are available from the Zenodo database (https://www.doi.org/10.5281/zenodo.7686233).

**Funding:** This work was partially funded by the National Council for Scientific and Technological Development (CNPq, Brazil). Erick G. Sperandio Nascimento is a CNPq technological development fellow (Proc. 308963/2022-9). The funder had no role in study design, data

## Abstract

Recent advancements in Artificial Intelligence have yielded promising results in addressing complex challenges within Natural Language Processing (NLP), serving as a vital tool for expediting judicial proceedings in the legal domain. This study focuses on the detection of similarity among judicial documents within an inference group, employing eight NLP techniques grounded in transformer architecture, specifically applied to a case study of legal proceedings in the Brazilian judicial system. The transformer-based models utilised — BERT, GPT-2, RoBERTa, and LlaMA — were pre-trained on general-purpose corpora of Brazilian Portuguese and subsequently fine-tuned for the legal sector using a dataset of 210,000 legal cases. Vector representations of each legal document were generated based on their embeddings, facilitating the clustering of lawsuits and enabling an evaluation of each model's performance through the cosine distance between group elements and their centroid. The results demonstrated that transformer-based models outperformed traditional NLP techniques, with the LlaMA model, specifically fine-tuned for the Brazilian legal domain, achieving the highest accuracy. This research presents a methodology employed in a real case involving substantial documentary content that can be adapted for various applications. It conducts a comparative analysis of existing techniques focused on a non-English language to quantitatively explain the results obtained with various NLP transformers-based models. This approach advances the current state of the art in NLP applications within the legal sector and contributes to the achievement of Sustainable Development Goals.

## Introduction

Recent advances in Natural Language Processing (NLP) have encouraged researchers to focus on techniques that transform short texts into vector representations, taking into account both

collection and analysis, decision to publish, or preparation of the manuscript.

**Competing interests:** The authors have declared that no competing interests exist.

the context and semantics of the words within the document. Studies have also demonstrated that machine learning algorithms are powerful tools for addressing complex problems in NLP [1]. To this end, several works can be highlighted that, considering word context, apply techniques for generating word embeddings, which are vector representations of words and, by extension, documents [2–5]. Additionally, recent studies have proposed adaptations of transformer models, including (i) Bidirectional Encoder Representations from Transformers (BERT) [6], (ii) Generative Pre-trained Transformer (GPT) [7], (iii) Robustly optimised BERT approach (RoBERTa) [8], and (iv) Large Language Model Meta AI (LlaMA) [9], all of which have demonstrated significant results. However, no study to date has consolidated a methodology that details the use of various NLP techniques, from traditional to cutting-edge approaches, applied to robust texts and tested in real-world cases. This reveals a fertile and unexplored opportunity in the legal sector to validate the methodology proposed in this work. Consequently, the use of word embeddings is crucial for analysing the large volumes of unstructured data typically presented in court.

Recent developments in the Brazilian Justice system have led to significant transformations, particularly with the digitisation of all procedural documents. In 2012, the Brazilian Labour Court implemented the Electronic Judicial Process (acronym in Portuguese for "Processo Judicial Eletrônico" - PJe) software system. Since then, all new lawsuits have been fully digital, and by 2023, 99.9% of cases in progress were being handled through this system [10].

Given the limitations of human beings in analysing large amounts of data within a reasonable time frame, especially when the data appears uncorrelated, it is possible to assist them in pattern recognition through data analysis, computational and statistical methods. As the volume of textual data continues to increase exponentially, analysing patterns in court documents is becoming increasingly challenging.

In this way, implementing actions aimed at speeding up the justice system can contribute to achieving the Sustainable Development Goals (SDGs) [11], particularly Goals 8 and 16, along with their respective targets, as summarised below:

- 8. Decent Work and Economic Growth - Promote inclusive and sustainable economic growth, employment and decent work for all:
  - 8.3 - Promote development-oriented policies that support productive activities, decent job creation, entrepreneurship, creativity and innovation, and encourage the formalisation and growth of micro-, small- and medium-sized enterprises, including through access to financial services;
  - 8.8 - Protect labour rights and promote safe and secure working environments for all workers, including migrant workers, in particular women migrants, and those in precarious employment.
- 16. Peace, Justice and Strong Institutions - Promote just, peaceful and inclusive societies:
  - 16.3 - Promote the rule of law at the national and international levels and ensure equal access to justice for all;
  - 16.6 - Develop effective, accountable and transparent institutions at all levels.

To optimise procedural progress, the Brazilian legal system employs several principles, such as procedural economy, efficiency, due process, and the reasonable duration of a case, to ensure the swift handling of judicial proceedings [12]. However, one of the major challenges for Brazilian Justice is meeting the growing judicial demand in a timely manner. Currently, specialists are responsible for triaging documents and distributing lawsuits to team members, which deviates from their primary task of drafting decisions. This situation contributes to an increase in the congestion rate (an indicator measuring the percentage of cases that remain

unresolved by the end of the year) and a decrease in the supply of demand index (acronym in Portuguese for "Índice de Atendimento à Demanda" - IAD - an indicator that measures the proportion of processed cases compared to new cases) [10].

The Brazilian labour process is inherently complex, posing challenges for legal professionals in organising structured information related to various judicial matters within the electronic judicial system. A significant hurdle lies in the absence of labelled data, as labelling in the legal sector demands extensive hours of work from experts, whose time is both very limited and costly. Consequently, the lack of prior expert classification demands employing pattern detection through grouping techniques. This approach aligns with one of the primary motivations of this work: to support legal advisors by facilitating the grouping of similar documents, thereby optimising their workflow. Moreover, the groups formed using the methodology proposed in this research could, in the future, streamline the efforts of experts in labelling data, ultimately enabling the application of supervised learning techniques.

By using a process grouping mechanism, it is possible to improve the allocation of work among advisers based on the similarity between the documents analysed. This method of distributing tasks can enhance efficiency in resolving pending cases, as advisers can focus on cases with similar requests, thereby optimising their workflow. Additionally, calculating similarity within these groupings aids in the search for relevant case law (A legal term meaning a set of previous judicial decisions following the same line of interpretation), ensuring that judgments adhere to the principle of legal certainty. According to Canotilho (2003) [13], the principle of legal certainty guarantees individuals the right to trust that judicial decisions are based on current and valid legal norms.

Therefore, to illustrate the potential benefits of grouping similar processes, Table 1 presents an estimated comparison between working methods with and without this grouping. It shows how the distribution of work among advisors and the search for previously judged similar legal cases can be optimised, emphasising the importance of this methodology.

This methodology, based on deep learning for grouping judicial processes, can be developed, tested, and deployed, particularly within the Brazilian Labour Court, through the tests and validations conducted. This research presents a methodology applied in a real case involving substantial documentary content, which can be adapted for various applications. Additionally, it conducts a comparative analysis of existing techniques focused on a non-English language to quantitatively explain the results obtained using various NLP transformer-based models.

**Table 1. A comparison estimate between work methods with and without the grouping of similar processes highlights the importance of this methodology.**

| Clustering | |
|---|---|
| **Without** | **With** |
| The chief advisor conducts a brief review of each process, spending **3 to 5 minutes per process**, to determine which advisor will handle it. | The chief advisor conducts a brief review of the grouped processes to determine which advisor will handle them, spending **3 to 5 minutes per group**, which contains several processes, thus allowing a much faster analysis of the documents. |
| The advisor thoroughly reviews each process and searches the database of previous decisions using keywords, "judging body", "collegian judging body", and others. | The advisor searches for previously judged cases within the group associated with the process for which he is preparing the draft decision to ensure consistency with prior rulings. |

This work aims to use the results discussed by Oliveira and Sperandio Nascimento (2021) [14] as a baseline, comparing them with the degree of similarity between judicial documents achieved in the inferred groups through unsupervised learning. This comparison involves the application of eight NLP techniques: (i) BERT trained for general purposes for Portuguese (BERT ptBR); (ii) BERT specialised with the corpus of Brazilian labour judiciary (BERT Jud.); (iii) GPT-2 trained for general purposes for Portuguese (GPT-2 ptBR); (iv) GPT-2 specialised with the corpus of Brazilian labour judiciary (GPT-2 Jud.); (v) RoBERTa trained for general purposes for Portuguese (RoBERTa ptBR); (vi) RoBERTa specialised with the corpus of Brazilian labour judiciary (RoBERTa Jud.); (vii) LlaMA 7B trained for general purposes for Portuguese (Sabiá 7B); and (viii) Sabiá 7B specialised with the corpus of Brazilian labour judiciary (Sabiá 7B Jud.). This methodology was tested on Brazilian labour legal documents, making it applicable to other areas of justice, both Brazilian and international, and potentially extendable to documents from other fields of knowledge.

Thus, as proposed by Oliveira and Sperandio Nascimento (2021) [14], the degree of similarity reflects the model's performance and results from the average similarity rate of the document groups. This rate is based on the cosine similarity between the elements of each group and its centroid, as well as the average cosine similarity among all documents within the group.

To delimit the scope of this research and enable a coherent comparison, the same dataset used in Oliveira and Sperandio Nascimento (2021) [14] was applied. The dataset includes approximately 210,000 Ordinary Appeals Filed (acronym in Portuguese for "Recurso Ordinário Interposto" - ROI) in legal proceedings (Available at: https://www.doi.org/10.5281/zenodo.7686233), representing about three years worth of documents related to Ordinary Appeals Filed in a medium-sized Brazilian Regional Labour Court. The Ordinary Appeal was chosen as a reference because it is typically the document that escalates a case to a higher court (2nd degree), resulting in the Ordinary Appeal (acronym in Portuguese for "Recurso Ordinário" - RO). This appeal serves as a free plea, an appropriate appeal to final and terminating judgments rendered at first instance, seeking a review by a hierarchically superior judicial body [15]. Therefore, Fig 1 is included to illustrate the macro flow of the labour process.

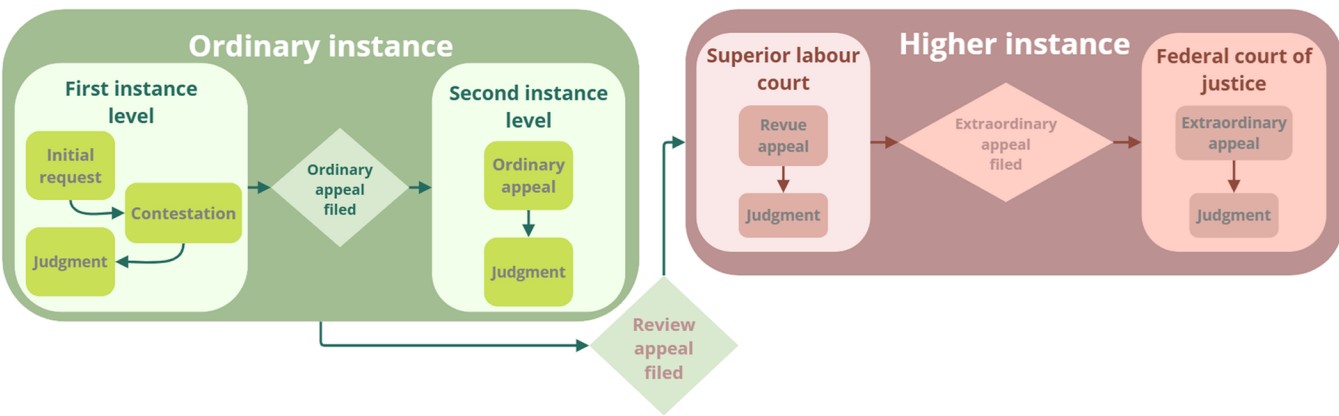

**Fig 1. Macro flow of the labour process.**

Therefore, the main innovations of this work are:

- This research presents a novel analysis of the applicability of transformer-based models for clustering similar documents in the legal field.
- Unsupervised learning was applied to identify patterns and determine the most effective and representative groups by clustering the embeddings generated by the transformer-based models for each legal document.
- A comprehensive methodology was developed to ensure the reproducibility of results, enabling its application to not only the legal sector but also other specialised domain applications, for both the tested models and future transformer-based models.
- The dataset used in this research has been made publicly available to support reproducibility and applicability to future studies in this field and beyond.

In this work, a literature review of unsupervised machine learning algorithms applied to the legal domain was conducted, focusing on the use of NLP. Additionally, recent techniques involving Artificial Intelligence (AI) algorithms for the generation of word embeddings are reviewed. Following the review, various methods were applied, and the results were compared. Finally, potential future challenges in the field are proposed.

## State-of-the-art review

Recent research maintains that machine learning algorithms hold great potential for solving high-complexity problems. These algorithms are categorised as (i) supervised, (ii) unsupervised, (iii) semi-supervised, and (iv) reinforcement learning [16]. In this study, a literature review was conducted to explore the most recent work published between 2017 and 2024. The review focused on unsupervised machine learning algorithms and clustering applied to the legal domain using NLP, and was carried out using the following databases: (i) Google Scholar, (ii) ScienceDirect, and (iii) IEEE Xplore.

The research revealed that, to date, few studies have addressed this topic, highlighting its complexity. Oliveira and Sperandio Nascimento (2021) [14] investigated the degree of similarity between judicial documents from the Brazilian Labour Court using unsupervised learning and various NLP techniques. These techniques included: (i) Term Frequency-Inverse Document Frequency (TF-IDF), (ii) Word2Vec with Continuous Bag of Words (CBoW), and (iii) Word2Vec with Skip-gram, both trained on general-purpose data for Brazilian Portuguese. Furthermore, Song et al. (2022) [17] conducted an empirical evaluation of pre-trained language models (PLMs) for legal NLP, assessing their effectiveness on a dataset comprising up to 57,000 documents.

Expanding research on the use of NLP in the judicial domain, a systematic review was conducted to explore the challenges faced in trial prediction systems. These systems aim to assist lawyers, judges, and civil servants by predicting case outcomes such as the probability of success or failure, sentencing durations, and applicable laws using deep learning models. The researchers provided a detailed review of empirical studies on legal judgment prediction, conceptual approaches to text classification methods, and an in-depth explanation of transformer models [18]. In Katz et al. (2023) [19], the authors emphasised that the field of Legal NLP is rapidly expanding in terms of research volume, linguistic diversity, and methodological sophistication. As a result, studies are increasingly tackling a wide range of tasks, pushing technical boundaries and successfully addressing more complex real-world challenges.

Therefore, to expand the scope of the research, the restriction to the legal domain was removed, which facilitated the identification of additional relevant publications. Renuka et al. (2021)[20] discussed the use of a content recommendation system that groups similar articles using k-means clustering, with document content vectorised through TF-IDF [21]. In D'Silva and Sharma (2020) [22], automatic text summarisation was performed using TF-IDF and k-means to group sentences from documents for the generation of summaries. The studies concluded that TF-IDF was the primary technique used for vectorising textual content and that k-means was the most commonly applied algorithm for unsupervised machine learning [21]. Additionally, research by Santana et al. (2022) [23] proposed a model based on Transformers for Portuguese, which generated word embeddings from texts published in a Brazilian newspaper. These texts were limited to 510 words and were used for news classification.

It is assumed that the selection of the most suitable technique for generating word embeddings requires extensive research, experimentation, and model comparison. Recent studies have demonstrated the effectiveness of word embeddings in enhancing the performance of AI algorithms for tasks such as pattern detection and classification. However, most of the reviewed research is based on a limited number of documents, with the content of these documents further restricted to a maximum of 510 words.

Mikolov et al. (2013) [3] proposed the Word2Vec Skip-gram and Continuous Bag of Words (CBoW) architectures, which were considered state-of-the-art at the time for calculating vector representations of words. Subsequently, Embeddings from Language Models (ELMo) [24], Flair [25], and context2vec [26]—libraries based on the Long Short-Term Memory Network (LSTM) [27]—introduced context-aware word embeddings. These models created distinct word embeddings for each occurrence of a word, enabling the capture of word meaning in context. LSTM models were widely applied in speech recognition, language modelling, sentiment analysis, and text prediction. Unlike Recurrent Neural Networks (RNNs), LSTMs have the ability to forget, remember, and update information, offering an improvement over traditional RNNs [28].

Since 2018, new techniques for generating word embeddings have emerged, including notable models such as (i) BERT [6], a context-sensitive model based on the Transformer architecture [29]; (ii) Sentence BERT (SBERT), a "Siamese" BERT model designed to enhance BERT's performance in measuring sentence similarity [30]; (iii) Text-to-Text Transfer Transformer (T5), a framework that treats NLP tasks as text-to-text problems, where both input and output are formatted as text [31]; (iv) GPT-2, a Transformer-based model with 1.5 billion parameters [7]; (v) RoBERTa, an improved version of BERT, trained for a longer time with more data [8]; and (vi) LlaMA, a collection of foundational language models ranging from 7 billion to 65 billion parameters [9].

Furthermore, few NLP models for the Portuguese language were found in the literature that include: (i) the BERT (large) model, trained on the brWaC corpus [32], which contains 2.7 billion tokens and was presented in the article "BERTimbau: Pretrained BERT Models for Brazilian Portuguese" [33]; (ii) the GPT-2 (small) model, trained on texts from Portuguese Wikipedia, and introduced in the article "GPorTuguese-2 (Portuguese GPT-2 Small): A Language Model for Portuguese Text Generation (and More NLP Tasks...)" [34]; (iii) the RoBERTa (base) model, trained on Portuguese Wikipedia texts, titled roberta-pt-br and published on Hugging Face [35]; and (iv) the LLaMA 7B model, trained on the Portuguese subset of the ClueWeb 2022 dataset [36,37], and presented in the article "Sabiá: Portuguese Large Language Models" [38].

This analysis facilitated advancements in the state of the art in NLP applied to the legal sector. A comparative study was conducted, implementing Transformer techniques such as

BERT, GPT-2, RoBERTa, and LlaMA, utilising both general-purpose models in Brazilian Portuguese (ptBR) and specialised models for the Labour Judiciary. Labour legal cases in Brazil were successfully clustered using the k-means algorithm and cosine similarity. Furthermore, this work contributed to the development of a validated methodology for the Brazilian Labour Judiciary, which is applicable across various fields of justice in Brazil and internationally, as well as in other domains of knowledge.

## Methodology

This section presents the protocol for reproducing the results and conducting a comparative analysis. The routines in this study were implemented using Python, along with the same libraries used in the study by Oliveira and Sperandio Nascimento (2021) [14].

The pipeline consists of the following phases: (i) data extraction, (ii) data cleaning, (iii) generation of word embedding models, (iv) calculation of document vector representation, (v) unsupervised learning, and (vi) calculation of similarity measures, as illustrated in Fig 2.

Phases (iii) and (iv) are detailed in the following sections, while the remaining phases are summarised below, as proposed by Oliveira and Sperandio Nascimento (2021) [14]:

- Data extraction: a dataset containing information from approximately 210,000 legal proceedings, specifically of the "Filed Ordinary Appeal" type (acronym in Portuguese for "Recurso Ordinário Interposto" — ROI), was extracted;
- Data cleaning: two forms of preprocessing were performed: (i) detecting the subjects of the Unified Procedural Table (acronym in Portuguese for "Tabela Processual Unificada" — TPU — Available at: https://www.tst.jus.br/web/corregedoria/tabelas-processuais) contained in the extracted documents; and (ii) cleaning the content of the documents using regular expressions, which involved removing HTML tags, replacing the names of individuals linked to the legal cases with the "tag" "parteprocesso" (party in the process), and replacing the names of judging bodies (e.g., "Tribunal Regional do Trabalho" [Regional Labour Court]) with the "tag" "orgaojulgador" (judging body), among other actions;
- Unsupervised learning: the k-means algorithm was adopted, and its implementation using cosine distance was shared on the author's GitHub account (Algorithm implementation with cosine distance: https://github.com/raphaelsouzaoliveira/kmeans);
- Calculation of the similarity measure: The cosine similarity measure was used to assess the quality of the inferred groups.

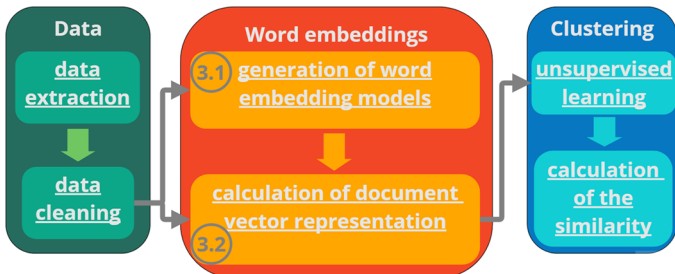

**Fig 2. Methodological flow.**

## Generation of word embedding models

The use of word vector representations, where numerical values capture relationships between words, is a crucial technique for solving machine learning problems involving textual data. In this research, word embeddings generated and shared for the Portuguese language were utilised, including: (i) BERTimbau; (ii) GPorTuguese-2 (Portuguese GPT-2 small); (iii) roberta-pt-br; and (iv) Sabiá 7B. In addition to these pre-trained Portuguese language models, recent literature suggests that embeddings tailored to the specific context of the problem may yield better results. Therefore, using the 210,000 extracted documents, four embedding generation techniques were applied: (i) fine-tuning the BERTimbau model; (ii) fine-tuning the GPorTuguese-2 model; (iii) fine-tuning the roberta-pt-br model; and (iv) fine-tuning the Sabiá 7B model, which will be detailed below.

**Fine-tuning of transformer models.** Recent studies demonstrate the benefits of applying transfer learning to generalist models, significantly enhancing results and achieving state-of-the-art performance in NLP [39]. For the fine-tuning of Transformer models, in addition to data cleaning, it is essential to adjust the data to maximise its benefits. Among the adjustments made, two deserve special attention: (i) definition of the sentence slot and (ii) implementation of a masking strategy (MASK) for the sentence tokens, which are detailed below.

Defining the sentence slot is a fundamental step in enabling the use of specialised data in the transfer learning process with a pre-trained model. Therefore, inspired by the strategy proposed in the article "Transformers: State-of-the-Art Natural Language Processing" [40], for each batch of 1,000 documents, all content is concatenated to create sentences of 128 tokens. If the last "sentence" in this batch contains fewer than 128 tokens, it is disregarded. Additional detailed approaches have been tested later, as presented in Fig 3.

To reduce the loss of context for words at the edges of sentences, the proposed approach, entitled Slot N/K, generates "sentences" with N tokens, as detailed below and illustrated in Fig 4.

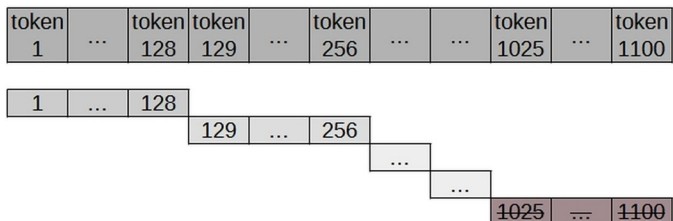

**Fig 3. Slot N - generation of "sentences" with 128 tokens.**

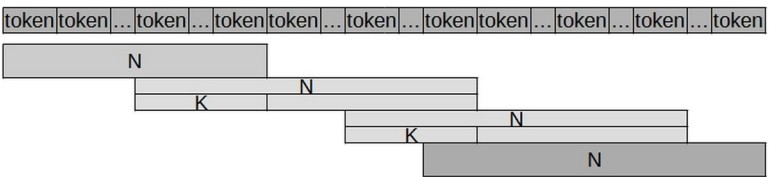

**Fig 4. Slot N/K generates "sentences" containing N tokens, where K represents the number of return tokens.**
These return tokens (K) are employed when the size of the "sentence" exceeds N, ensuring that words at the edges retain semantic context.

- Initial Slot: "sentence" formed by the first N tokens;
- Intermediate slots: "sentence" formed by N tokens counted from the (N-K) token of the previous "sentence", where K is the number of return tokens;
- Final Slot: "sentence" formed by the last N tokens.

Based on the approach detailed above, simulations were performed using the BERT model with the following settings: (i) Slot 128/16; (ii) Slot 128/32; (iii) Slot 128/64; (iv) Slot 256/64; (v) Slot 512/64; and (vi) Slot 64/16. These settings were compared with each other and with the approach proposed by Wolf et al. (2020) [40]. Among these, the Slot 128/32 setting achieved the best performance in specialising the BERT Transformer model for Portuguese using the judicial corpus (Fig 5). However, to optimise the architecture of each type of Transformer model, N was empirically defined as the maximum number of tokens supported by each architecture, while K was set as the integer value of 10% of N.

In the learning process, depending on the Transformers model used, a token masking strategy is applied to each sentence. Masked Language Models (MLM) are utilised for BERT and RoBERTa models, while Causal Language Models (CLM) are employed for GPT-2 and LlaMA models. The CLM is trained unidirectionally to predict the next word based on the preceding words [41], whereas the MLM employs a bidirectional approach to predict the masked words in the sentence.

Hence, for the training process of BERT models, simulations inspired by the article "Transformers: State-of-the-Art Natural Language Processing" [40] were conducted using masking rates of 15% and 25%. The results indicated that the 15% masking rate yielded the best performance in specialising the BERT model for Portuguese with the judicial corpus. Consequently, this masking rate was also applied to the RoBERTa model.

Finally, it is worth highlighting that it is essential to fine-tune the vocabulary using a specialised corpus. This process ensures that the model can incorporate new context more effectively, thereby enhancing its understanding of key terms. Additionally, fine-tuning helps the model grasp the specific language and jargon related to the content it will handle in NLP tasks.

Thus, as shown in Table 2, the fine-tuning of each model for the legal domain adhered to the same hyperparameters, tokenization techniques, and masking methods used during the training of their base models.

Thus, after fine-tuning the models to create specialised NLP models for the legal field, as well as adapting pre-existing models in Portuguese, it was necessary to calculate the vector

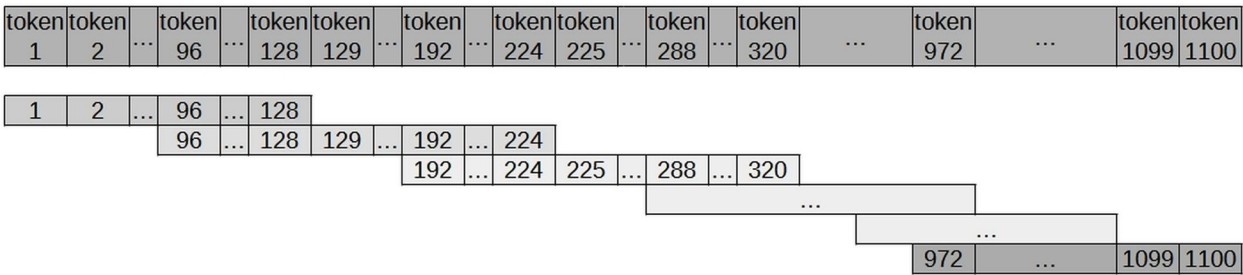

**Fig 5. Slot 128/32 generates "sentences" containing 128 tokens.** For sentences exceeding 128 tokens, a return of 32 tokens is applied to preserve semantic context.

**Table 2. Fine-tuning process details for each transformer architecture.**

| Model | Hiperparameters | Tokenization type | Masking method |
|---|---|---|---|
| BERT Jud. | 24-layer, 1024-hidden, | | |
| | 16-heads, 340M parameters | word-piece | MLM |
| GPT-2 Jud. | 12-layer, 768-hidden, | | |
| | 12-heads, 117M parameters | byte-level BPE | CLM |
| RoBERTa Jud. | 12-layer, 768-hidden, | | |
| | 12-heads, 125M parameters | byte-level BPE | MLM |
| Sabiá 7B Jud. | 32-layer, 4096-hidden, | | |
| | 32-heads, 6.7B parameters | sentence-piece BPE | CLM |

representations of the documents. This process is detailed in the following section. Subsequently, document embeddings for each of the eight models were used in an unsupervised training technique to generate the clusters.

## Calculation of document vector representation

Vector representation techniques for words, such as (i) BERT, (ii) GPT-2, (iii) RoBERTa, and (iv) LlaMA, must be utilised to calculate document embeddings from their respective word embeddings.

It is initially essential to detail the process of obtaining word embeddings for Transformer techniques. One advantage of these techniques over earlier methods, such as Word2Vec, is their ability to capture the vector representation of a word according to its global context. This means that the same word can have multiple vector representations depending on its usage. For example, consider the word "bank" (translated as *banco* in Portuguese) in the following two sentences: (i) "I go to the bank to withdraw money" ("Vou ao *banco* para sacar dinheiro" in Portuguese) and (ii) "I will sit on the bench of the square" ("Eu vou sentar no *banco* da praça" in Portuguese). In this case, Word2Vec would produce a unique vector representation for "*banco*", regardless of the context, while BERT, GPT-2, RoBERTa, and LlaMA would generate different embeddings based on the surrounding words.

Therefore, for Transformer models, it is necessary to "divide" the entire document into "slots" of sentences (context windows). Each model type has a limit on the number of accepted tokens per sentence and requires that the first and last tokens be special, namely [CLS] and [SEP]. Consequently, the slot sizes have been set as follows: (i) 510 tokens per sentence for the BERT and RoBERTa models; (ii) 1022 tokens per sentence for the GPT-2 model; and (iii) 2046 tokens per sentence for the LlaMA model.

As indicated in section State-of-the-art review, unlike the present work, which utilises very large documents, current research typically limits the texts used in vector transformations to the maximum number of words supported by the respective architecture. Fig 6 shows a histogram and distribution plot, highlighting that most documents in the dataset contain up to approximately 3,500 words, with an average of 2,525 words. Additionally, documents exceeding 10,000 words are considered outliers in the distribution, totalling 13661 documents. Despite this they were still considered in this research. It is important to note that the models considered in this research have input size limits ranging from 512 tokens (BERT) to 4096 tokens (Sabiá 7B). Therefore, the approach used in this study was crucial to enable the models to process more tokens than their maximum input size at one time.

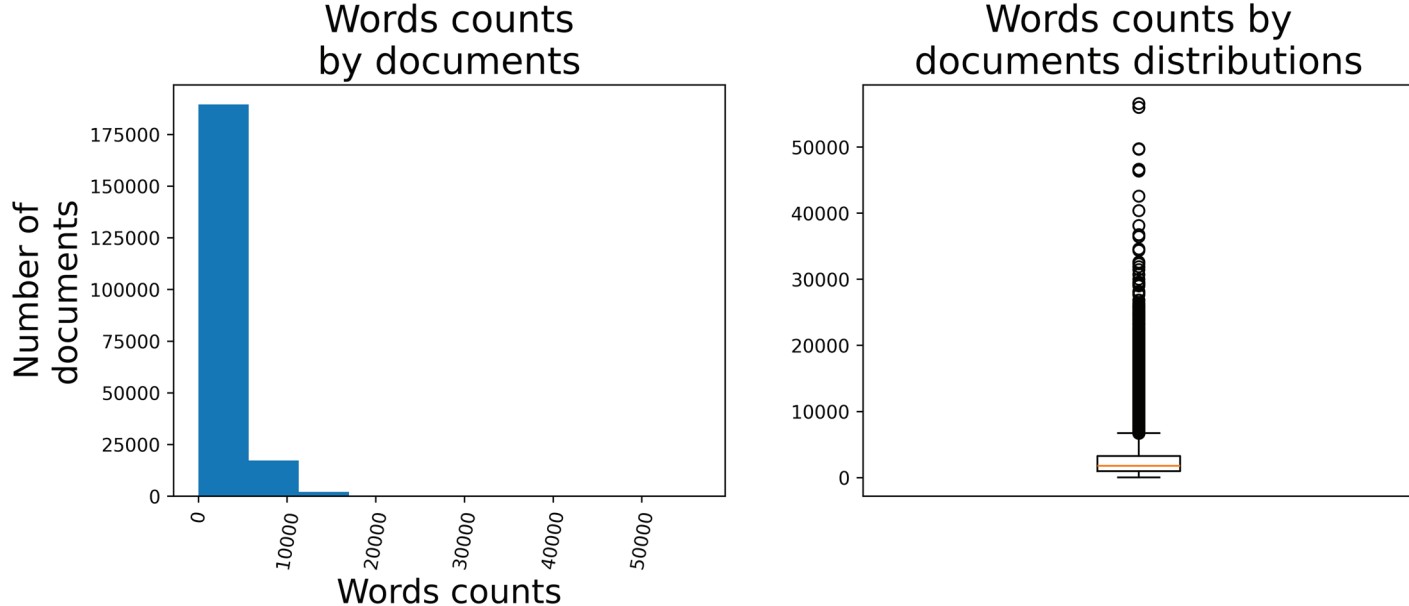

**Fig 6. Histogram of Words count by documents and Words count by documents distribution graphs.**

To address this limitation, strategies were developed to leverage all the word embeddings of a document while ensuring that the generated sentences retain their contextual integrity. These approaches, as illustrated in Fig 4, involve constructing sentences with N tokens, as described below.

- Initial Sentence: a "sentence" formed by the first N tokens;
- Intermediate Sentences: a "sentence" consisting of N tokens, starting from token N - K of the previous "sentence", where K is empirically set to 10% of N;
- Final Sentence: a "sentence" formed by the last N tokens.

Therefore, the sentences generated from each document include overlapping tokens selected to improve adherence to the context within the text. To this end, two approaches were tested: (i) averaging the word embeddings of overlapping tokens, and (ii) using the first K/2 overlapping tokens from the previous sentence combined with the last K/2 overlapping tokens from the current sentence. The latter approach yielded superior results in our simulations.

Hence, as shown in Fig 7, where N=510 and K=64, the overlapping tokens between the current and previous sentences are utilised as follows: (i) the first 32 overlapping tokens from the previous sentence (for example, tokens 446 to 477 from Slot 1); and (ii) the last 32 overlapping tokens from the current sentence (for example, tokens 478 to 510 from Slot 2).

Following the generation of word embeddings, the technique used by Oliveira and Sperandio Nascimento (2021) [14] was adopted to compute document embeddings. Specifically, this involved calculating the average of the word embeddings across all words in the document, with each embedding weighted according to its TF-IDF value.

Therefore, to provide a comprehensive overview, below summarises the parameters used for training the eight models utilised in this research.

| token 1 | token 2 | ... | token 446 | ... | token 477 | token 478 | ... | token 510 | token 511 | ... | token 590 | ... | token 773 | token 774 | ... | token 956 | token 957 | ... | ... | token 1099 | token 1100 |
|---|---|---|---|---|---|---|---|---|---|---|---|---|---|---|---|---|---|---|---|---|---|

| Slot 1 | | | | | | | | |
|---|---|---|---|---|---|---|---|---|
| 1 | 2 | ... | 446 | ... | 477 | 478 | ... | 510 |

| Slot 2 | | | | | | | | | | |
|---|---|---|---|---|---|---|---|---|---|---|
| 446 | ... | 477 | 478 | ... | 510 | 511 | ... | 590 | ... | 773 | 774 | ... | 956 |

| Slot 3 | | | | | | | | |
|---|---|---|---|---|---|---|---|---|
| 590 | ... | 773 | 774 | ... | 956 | 957 | ... | ... | 1099 | 1100 |

| Slot 1 | | | | | | Slot 2 | | | | | | Slot 3 | | | | | |
|---|---|---|---|---|---|---|---|---|---|---|---|---|---|---|---|---|---|
| 1 | 2 | ... | 446 | ... | 477 | 478 | ... | 510 | 511 | ... | 590 | ... | 773 | 774 | ... | 956 | 957 | ... | ... | 1099 | 1100 |

**Fig 7. A word embedding generation strategy is employed, demonstrated with an example where N = 510 and k = 64.** In this case, the overlapping tokens between the current and previous sentences are utilized within a 32-by-32 base to ensure continuity.

- BERT:
  - Tokenization type: word-piece;
  - Model details: 24-layer, 1024-hidden, 16-heads, 340M parameters;
  - Token mask type: Masked;
  - Data for training:
    - BERTimbau: brWac corpus;
    - BERT Jud.: 210K ROIs;
- GPT-2:
  - Tokenization type: byte-level BPE;
  - Model details: 12-layer, 768-hidden, 12-heads, 117M parameters;
  - Token mask type: Causal;
  - Data for training:
    - GPortuguese-2: Wikipedia in Portuguese;
    - GPT-2 Jud.: 210K ROIs;
- RoBERTa: Tokenization type: Model details: Token mask type: .
  - Tokenization type: byte-level BPE;
  - Model details: 12-layer, 768-hidden, 12-heads, 125M parameters;
  - Token mask type: Masked;
  - Data for training:
    - roberta-pt-br: Wikipedia in Portuguese;
    - RoBERTa Jud.: 210K ROIs;
- LlaMA 7B:
  - Tokenization type: sentence-piece BPE;
  - Model details: 32-layer, 4096-hidden, 32-heads, 6.7B parameters;
  - Token mask type: Causal;
  - Data for training:
    - Sabiá 7B: Portuguese subset of the ClueWeb 2022 dataset;
    - Sabiá 7B Jud.: 210K ROIs;

It is also worth noting that court documents typically contain a large number of words. NLP models with larger context windows, capable of handling more tokens at once, can easily accommodate entire documents in a single input sequence. This allows these models to fully capture the document's context without requiring text segmentation, unlike models

with smaller context windows, as demonstrated in this research. When implementing this methodology, it is important to consider the size of the model's context window and determine whether text segmentation is necessary. In cases where the context window is smaller than the number of words in the document, the text segmentation technique proposed in this research is particularly appropriate.

Moreover, after generating the unsupervised machine learning model and calculating the similarity measure, as defined by Oliveira and Sperandio Nascimento (2021) [14], a two-dimensional graphic representation of the vectorised documents was produced using the T-Distributed Stochastic Neighbour Embedding (t-SNE) reduction technique. This technique minimises divergence between two distributions by comparing the similarities between pairs of input objects and the corresponding low-dimensional points in the embedding [42].

## Results and discussion

By applying the previously detailed methodology, this research demonstrates how NLP techniques, combined with machine learning algorithms, are essential for optimising the operational costs of judicial processes, such as document screening and procedural distribution. These technologies improve time management by allowing experts to focus on their core activities. Moreover, this contributes to building a more sustainable society by supporting the achievement of United Nations Sustainable Development Goals 8 and 16: Decent Work and Economic Growth, and Peace, Justice, and Strong Institutions.

To apply the k-means unsupervised learning algorithm, it was necessary to define the optimal value of K for clustering. Inertia was used as a metric to assess how well the data were grouped by k-means. Inertia is calculated as the sum of the squared distances between each point and its centroid, with the goal of minimising inertia. As K increases, inertia typically decreases. Therefore, the elbow method was applied to identify the point at which the rate of inertia reduction begins to slow down. Thirty-one values for K were tested, ranging from 30 to 61 in unit intervals, and the optimal K was selected based on the elbow method, as shown in Fig 8.

The analysis of the inertia plots presented in Fig 8 reveals that, for all models except Sabiá 7B Jud., the inertia generally decreased as the number of clusters increased, although with significant variance. An asymptotic downward trend can be observed through the dashed lines in these plots. Furthermore, the results emphasise that the inertia values obtained for all specialised models (BERT Jud., GPT-2 Jud., RoBERTa Jud., and Sabiá 7B Jud.) are significantly lower than those of their general-purpose counterparts. This finding underscores the value of domain-specific specialisation in improving clustering performance, as it leads to better-defined clusters. Notably, for the Sabiá 7B Jud. model, the analysis clearly indicates that the optimal number of clusters is achieved with k=55.

After determining the optimal K, the k-means model was trained, and the resulting clusters were used to calculate: (i) the average similarity between the documents within each group, providing an overview of how the documents were distributed across the clusters generated by each NLP technique; and (ii) the mean similarity between the documents and their respective centroids, which helped identify the technique with the best performance.

To demonstrate the advancements introduced by this research, Table 3 presents results extracted from Oliveira and Sperandio Nascimento (2021) [14], which established a baseline for applying NLP techniques in the legal field for similar purposes. Notably, the Word2Vec Skip-gram ptBR technique was highlighted in that study as the most effective method for generating word embeddings aimed at clustering judicial documents of the Filed Ordinary Appeal type.

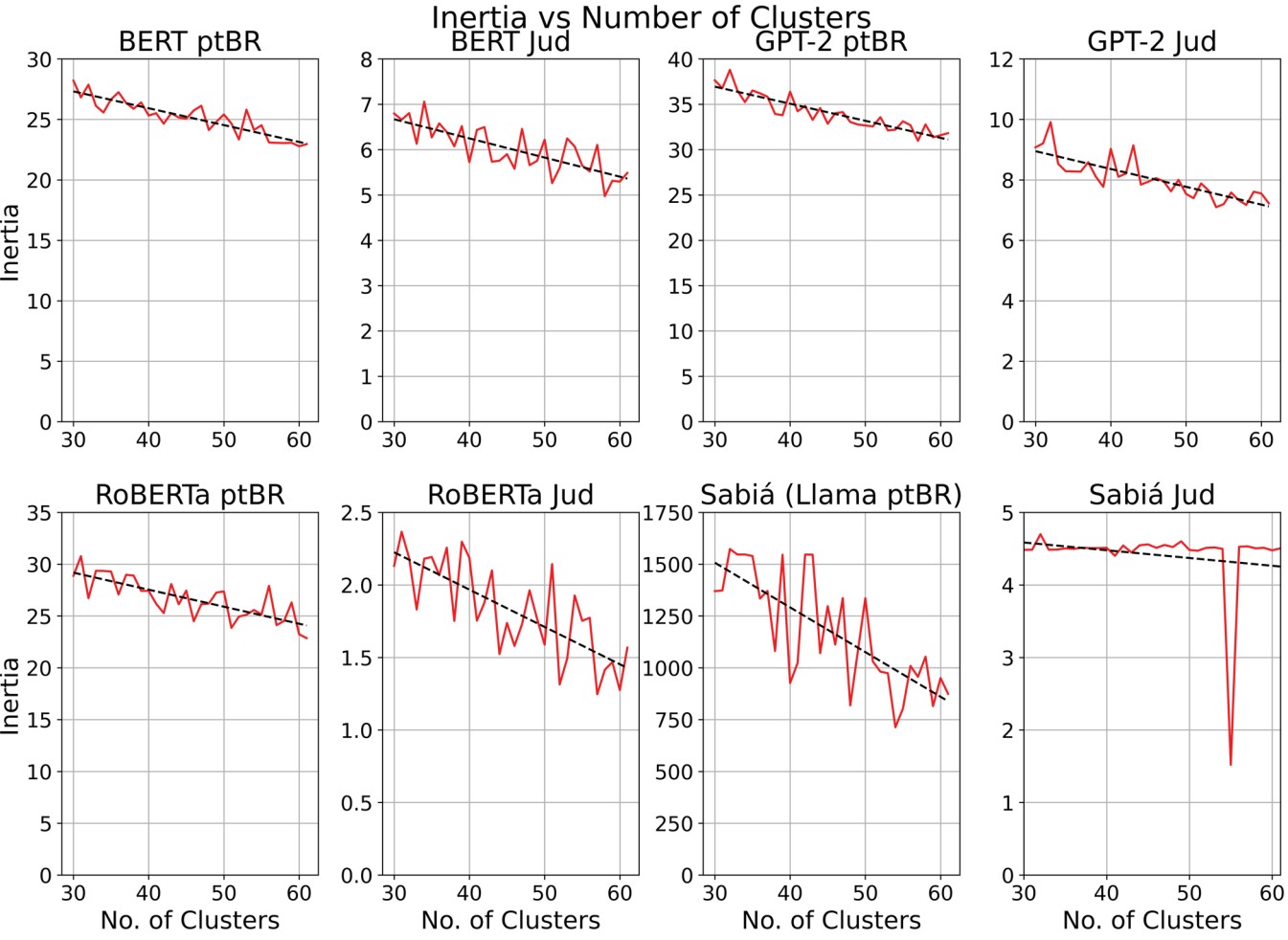

**Fig 8. Inertia charts were constructed using the elbow method to determine the optimal number of clusters for each approach.**

The statistical data for the average similarity between documents within each group and the average similarity between group documents and their centroids, as shown in Table 4 and Table 5, along with the comparative distribution charts (Figs 9 and 10), demonstrate that transformer models with fewer parameters (BERT, GPT-2, and RoBERTa) generally performed better when using word embeddings generated from specialised legal corpora compared to those based on general-purpose Portuguese (ptBR) corpora. Notably, the specialised

**Table 3. Statistical data extracted from the work "Clustering by Similarity of Brazilian Legal Documents Using Natural Language Processing Approaches" [14]. Subtitle: TF-IDF - Term Frequency - Inverse Document Frequency; W2Vc - Word2Vec CBow ptBR; W2Vsg - Word2Vec Skip-gram ptBR; Grp - number of groups formed; Mean - mean of the values; Std - standard deviation; Min - minimum of the values; $Q_1$ - first quartile; $Q_2$ - second quartile; $Q_3$ - third quartile; Max - maximum of the values.**

| Type | Grp | Mean | Std | Min | $Q_1$ | $Q_2$ | $Q_3$ | Max |
|------|-----|------|-----|-----|-------|-------|-------|-----|
| TF-IDF | 49 | 0.624 | 0.172 | 0.247 | 0.502 | 0.586 | 0.164 | 0.964 |
| W2Vc | 59 | 0.947 | 0.063 | 0.764 | **0.935** | **0.979** | 0.991 | 0.999 |
| W2Vsg | **34** | **0.948** | **0.061** | **0.796** | 0.925 | 0.976 | **0.992** | **0.999** |

models produced closely comparable results, particularly between the BERT Jud. and GPT-2 Jud. models. For models with more parameters (Sabiá 7B and Sabiá 7B Jud.), the generalist Sabiá 7B performed poorly, likely due to the tokenizer not being specialised before finetuning the LlaMA model with Portuguese data [38]. In contrast, the specialisation of the tokenizer before fine-tuning may have been key to the Sabiá 7B Jud. model achieving the best overall performance.

When comparing the values presented in Table 4 and Table 5, it is noteworthy that the results in Table 4 are consistently slightly lower. This suggests that the similarity measurement shown in Table 4 might reduce the overall similarity rate due to the possibility of certain elements being positioned in opposing parts of the group. Furthermore, from Figs 9 and 10, it is evident that the groupings generated by all techniques are highly cohesive, particularly in the case of the specialised techniques. In transformer models with fewer parameters, the expert models produced fewer outliers compared to the generalist techniques, indicating more consistent groupings.

To ensure a fair comparison of the number of groups formed, the K obtained by the Sabiá 7B Jud. technique was established as the standard, as it achieved the best result. Consequently, the grouping with K=55 was used for all techniques, as presented in Table 6. This approach

**Table 4. Statistics of the cosine similarity between all elements of the group, where the ptBR models are generalist and the Jud. models are specialised. The best results are highlighted in bold. <u>Subtitle</u>: Bbr - BERT ptBR; Bj - BERT Jud.; Gbr - GPT-2 ptBR; Gj - GPT-2 Jud.; Rbr - RoBERTa ptBR; Rj - RoBERTa Jud.; Sbr - Sabiá 7B ptBR; Sj - Sabiá 7B Jud.; Grp - number of groups formed; Mean - mean of the values; Std - standard deviation; Min - minimum of the values; $Q_1$ - first quartile; $Q_2$ - second quartile; $Q_3$ - third quartile; Max - maximum of the values.**

| Model | Grp | Mean | Std | Min | $Q_1$ | $Q_2$ | $Q_3$ | Max |
|---|---|---|---|---|---|---|---|---|
| Bbr | 34 | 0.975 | 0.018 | 0.897 | 0.971 | 0.980 | 0.985 | 0.992 |
| Bj | 40 | 0.989 | 0.007 | 0.967 | 0.987 | 0.991 | 0.994 | 0.997 |
| Gbr | 38 | 0.974 | 0.018 | 0.933 | 0.966 | 0.980 | 0.986 | 0.995 |
| Gj | **32** | 0.989 | **0.007** | **0.967** | 0.988 | 0.991 | 0.993 | 0.998 |
| Rbr | 56 | 0.981 | 0.011 | 0.935 | 0.977 | 0.984 | 0.989 | 0.996 |
| Rj | 52 | 0.993 | 0.017 | 0.874 | 0.995 | 0.997 | 0.998 | 0.999 |
| Sbr | 113 | 0.727 | 0.253 | 0.000 | 0.529 | 0.768 | 0.995 | 1.000 |
| Sj | 55 | **0.996** | 0.016 | 0.899 | **0.999** | **1.000** | **1.000** | **1.000** |

**Table 5. Statistics of the cosine similarity of the group elements to the centroids, where the ptBR models are generalist and the Jud. models are specialised. The best results are highlighted in bold. <u>Subtitle</u>: Bbr - BERT ptBR; Bj - BERT Jud.; Gbr - GPT-2 ptBR; Gj - GPT-2 Jud.; Rbr - RoBERTa ptBR; Rj - RoBERTa Jud.; Sbr - Sabiá 7B ptBR; Sj - Sabiá 7B Jud.; Grp - number of groups formed; Mean - mean of the values; Std - standard deviation; Min - minimum of the values; $Q_1$ - first quartile; $Q_2$ - second quartile; $Q_3$ - third quartile; Max - maximum of the values.**

| Model | Grp | Mean | Std | Min | $Q_1$ | $Q_2$ | $Q_3$ | Max |
|---|---|---|---|---|---|---|---|---|
| Bbr | 34 | 0.987 | 0.009 | 0.947 | 0.985 | 0.990 | 0.993 | 0.996 |
| Bj | 40 | 0.995 | 0.003 | 0.983 | 0.994 | 0.995 | 0.997 | 0.999 |
| Gbr | 38 | 0.997 | 0.009 | 0.966 | 0.983 | 0.990 | 0.993 | 0.997 |
| Gj | **32** | 0.995 | **0.003** | **0.983** | 0.994 | 0.996 | 0.997 | 0.999 |
| Rbr | 56 | 0.990 | 0.006 | 0.967 | 0.989 | 0.992 | 0.995 | 0.998 |
| Rj | 52 | 0.997 | 0.009 | 0.937 | 0.998 | 0.998 | 0.999 | 0.999 |
| Sbr | 113 | 0.855 | 0.140 | 0.563 | 0.746 | 0.886 | 0.998 | 1.000 |
| Sj | 55 | **0.998** | 0.008 | 0.949 | **1.000** | **1.000** | **1.000** | **1.000** |

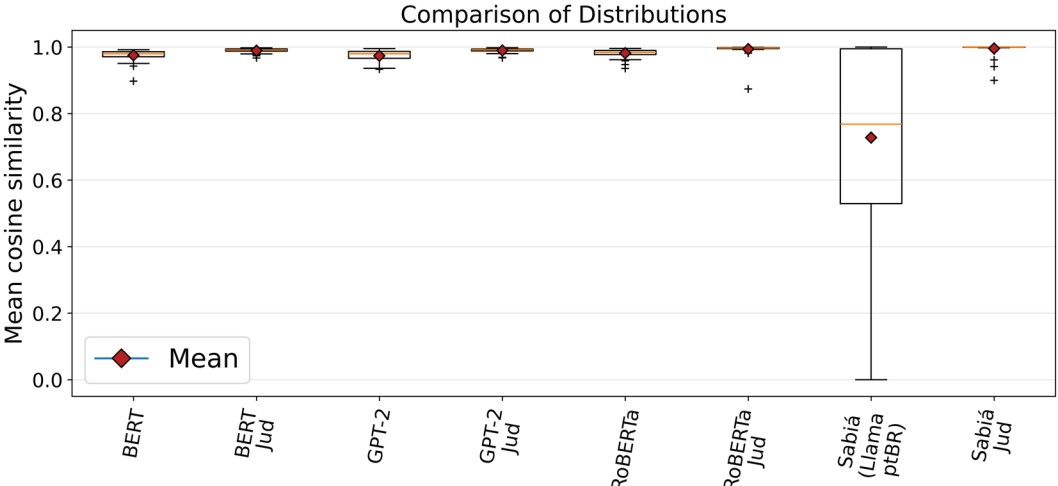

**Fig 9. Comparison chart of the distribution of the average similarity between the group documents.** A more cohesive distribution (narrower boxes) with fewer outliers indicates better performance.

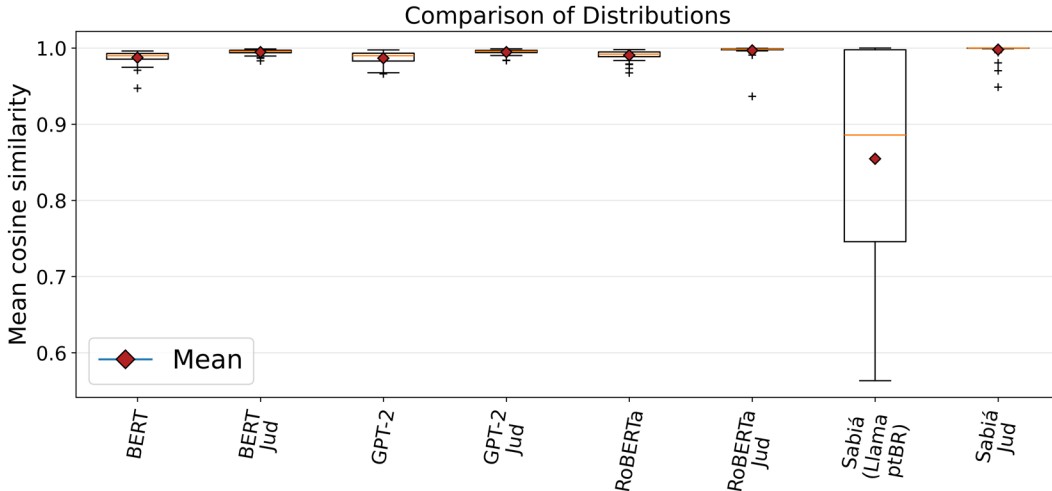

**Fig 10. Comparison chart of the distribution of the average similarity between group documents and their centroid.** A more cohesive distribution (narrower boxes) with fewer outliers indicates better performance.

consolidated the Sabiá 7B Jud. model as the best for grouping legal documents using embeddings from transformer-based language models, ensuring consistency with the results presented earlier in Table 5.

Given that most techniques produced similar results, it is important to present the processing time for each Transformer technique using a computer with 40 physical cores and 196 GB of memory. This setup was utilised to generate numerical representations for approximately 210,000 judicial documents of the Filed Ordinary Appeal type. As shown in Table 7, GPT-2 achieved a significantly higher average vectorisation rate per minute compared to BERT. However, as expected, RoBERTa outperformed both BERT and GPT-2. According to Liu et al. (2019) [8], RoBERTa's performance can be enhanced by training for longer periods with larger batch sizes and more data, while avoiding the next-sentence prediction strategy, and

**Table 6. Statistics of the cosine similarity of the elements of the 55 groups with their centroids, where the ptBR models are generalists and the Jud. models are specialised. The best results are highlighted in bold. <u>Subtitle</u>: Bbr - BERT ptBR; Bj - BERT Jud.; Gbr - GPT-2 ptBR; Gj - GPT-2 Jud.; Rbr - RoBERTa ptBR; Rj - RoBERTa Jud.; Sbr - Sabiá 7B ptBR; Sj - Sabiá 7B Jud.; Grp - number of groups formed; Mean - mean of the values; Std - standard deviation; Min - minimum of the values; $Q_1$ - first quartile; $Q_2$ - second quartile; $Q_3$ - third quartile; Max - maximum of the values.**

| Model | Mean | Std | Min | $Q_1$ | $Q_2$ | $Q_3$ | Max |
|---|---|---|---|---|---|---|---|
| Bbr | 0.989 | 0.007 | 0.948 | 0.987 | 0.991 | 0.994 | 0.998 |
| Bj | 0.995 | **0.003** | **0.984** | 0.994 | 0.996 | 0.997 | 0.999 |
| Gbr | 0.988 | 0.008 | 0.963 | 0.987 | 0.990 | 0.993 | 0.998 |
| Gj | 0.995 | 0.004 | 0.979 | 0.994 | 0.996 | 0.997 | 0.998 |
| Rbr | 0.990 | 0.007 | 0.962 | 0.989 | 0.992 | 0.995 | 0.999 |
| Rj | 0.997 | 0.008 | 0.937 | 0.998 | 0.999 | 0.999 | 1.000 |
| Sbr | 0.850 | 0.135 | 0.532 | 0.744 | 0.866 | 0.997 | 1.000 |
| Sj | **0.998** | 0.008 | 0.949 | **1.000** | **1.000** | **1.000** | **1.000** |

**Table 7. Average processed documents per minute on CPUs for each model, with the best result highlighted in bold.**

| Transformer Model | Average document processing rate per minute on CPUs |
|---|---|
| BERT ptBR | 6.45 |
| BERT Jud. | 9.62 |
| GPT-2 ptBR | 29.40 |
| GPT-2 Jud. | 29.03 |
| RoBERTa ptBR | **55.31** |
| RoBERTa Jud. | 53.73 |
| Sabiá 7B | 2.25 |
| Sabiá 7B Jud. | 2.23 |

by training longer sequences with dynamically altered masking. The LlaMA models (Sabiá 7B and Sabiá 7B Jud.) exhibited the worst performance, making document vectorisation impractical on CPUs. These models required the use of 4 A100 GPU nodes with 40 GB of memory each, which increased the vectorisation rate to 66.66 documents per minute. Thus, performance evaluation is crucial when dealing with large quantities of documents containing extensive content, further highlighting the significance of this research.

Given the above, among all the techniques evaluated, the Sabiá 7B Jud. technique was the best option for generating word embeddings for clustering judicial documents of the Filed Ordinary Appeal type. However, it also had the longest processing time, necessitating the use of GPUs to complete the task within an acceptable time frame. On the other hand, when considering both the processing time and the final metric results, the RoBERTa Jud. technique could be regarded as the optimal choice for generating word embeddings for clustering judicial documents of the Filed Ordinary Appeal type.

It is important to emphasise that the results of this research (Table 5) demonstrate significant advancements compared to the findings presented in previous research (Table 3). In that earlier study, the best average cosine similarity of the elements within the group to the centroid was 0.948, while this research achieved a similarity of 0.998. This represents an improvement of 5 percentage points through the use of Transformer architecture. Thus, this advancement suggests that the techniques outlined in this methodology may provide a solid foundation for application in other areas of Brazilian and international justice, as well as in various fields of knowledge.

One noteworthy finding in the results is that specialised word embedding techniques consistently outperformed their general counterparts across all transformer models. This supports the hypothesis that model specialisation leads to improved outcomes, even when the corpus used for specialisation is significantly smaller than the corpus used to train the foundational model.

The results achieved by each approach can be visualised in a two-dimensional projection of the groups formed across the eight techniques (available in the supporting information): (i) BERT ptBR (S1 Fig); (ii) BERT Jud. (S2 Fig); (iii) GPT-2 ptBR (S3 Fig); (iv) GPT-2 Jud.

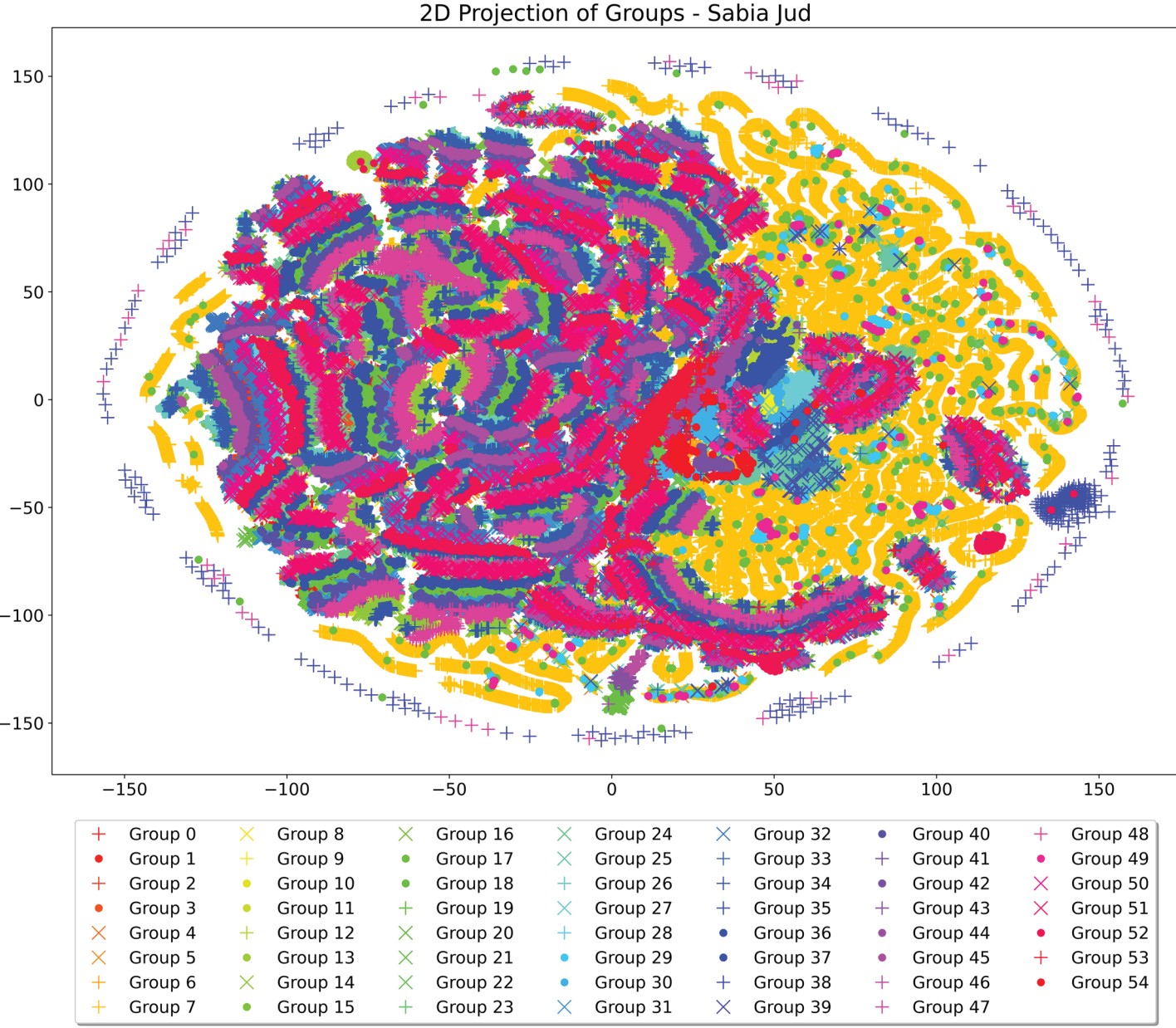

**Fig 11. Groups of documents formed using the Sabiá 7B Jud. technique, projected in two dimensions based on the test dataset.**

(S4 Fig); (v) RoBERTa ptBR (S5 Fig); (vi) RoBERTa Jud. (S6 Fig); (vii) Sabiá 7B (S7 Fig); and (viii) Sabiá 7B Jud. (Fig 11). A qualitative analysis reveals that the groups formed using Sabiá 7B Jud. (Fig 11) are notably better defined, corroborating the previously discussed findings. Moreover, visualising the two-dimensional projections of the best technique from Oliveira and Sperandio Nascimento (2021) [14] alongside the current work (Fig 11) reinforces the significance of establishing a methodology for pattern detection using NLP applied to large volumes of complex documents.

To assist legal advisors in better identifying which TPUs are most related to each group, a simple and effective method for determining these terms was proposed. To provide semantic meaning to the groups, the most significant labels were assigned to the 55 groups. This process involved extracting the three most relevant TPU terms from the documents within each group and using them as labels. For the technique that achieved the best performance, Sabiá 7B Jud., embeddings of these TPU terms were calculated, and their cosine similarity to the group centroid was determined. The three terms with the highest cosine similarity were selected to name each group. Table 8 presents the group names defined by this technique for the groups with the best performance.

Based on Table 8, the high similarity between the most relevant terms and the centroid for each group stands out. This reinforces the quality of the groups generated by the technique, particularly in the context of the Labour Court, where this type of documents is typically associated with distinct terms. Additionally, it is noticed that in some groups, such as 8 and 47, the three terms are repeated, but in a different order. This means that, although their

**Table 8. TPU terms most relevant to groups with average similarity greater than 0.999999, ordered by cosine similarity.**

| Group id | Amount of documents | Average similarity | Group name TPUs (Cosine similarity) |
|---|---|---|---|
| 7 | 49,638 | 1.000000 | ctps (0.999804), período de graça (0.999796), astreintes (0.999795) |
| 8 | 2,689 | 0.999999 | licença previdenciária (0.999999), ação de cobrança (0.999999), descontos previdenciários (0.999998) |
| 10 | 3,861 | 0.999999 | licença previdenciária (0.999999), suspeição (0.999998), descontos previdenciários (0.999998) |
| 13 | 6,284 | 0.999999 | agravo de instrumento (0.999999), cesta básica (0.999999), compensação em atividade insalubre (0.999999) |
| 16 | 3,785 | 0.999999 | ente público (0.999999), dirigente sindical (0.999999), cálculo repercussão (0.999999) |
| 21 | 5,793 | 0.999999 | seguro de vida (0.999998), atraso na audiência (0.999997), sucessão de empregadores (0.999997) |
| 23 | 5,947 | 0.999999 | suspeição (0.999999), licença previdenciária (0.999998), deficiente físico (0.999998) |
| 24 | 4,962 | 0.999999 | direito de greve (0.999999), remuneração, verbas indenizatórias e benefícios (0.999999), ação de cobrança (0.999999) |
| 26 | 2,709 | 0.999999 | promoção (0.999999), bancos (0.999999), perícia (0.999999) |
| 28 | 4,825 | 0.999999 | ação cautelar (0.999999), compensação em atividade insalubre (0.999999), tomador de serviços terceirização (0.999998) |
| 34 | 2,788 | 0.999999 | salário família (0.999999), salário in natura (0.999999), extinção normal do contrato a termo (0.999998) |
| 38 | 4,201 | 0.999999 | deficiente físico (0.999999), engenheiro, arquiteto e engenheiro agrônomo (0.999999), agravo de instrumento (0.999999) |
| 47 | 5,250 | 0.999999 | descontos previdenciários (0.999999), licença previdenciária (0.999999), ação de cobrança (0.999998) |

similarity levels are very close, they hold different semantic significance, which is important for decision-making processes taken by legal assessors during procedural evaluations. The full relation of groups and their most relevant TPUs based on this analysis can be found in Supporting information S1 Table.

Based on the methodology developed and evaluated in this work, a tool named GEMINI was created for the Brazilian Labour Court. This tool aids in searching for jurisprudence, distributing work among experts, and identifying opportunities to standardise legal interpretations within the courts, thereby establishing Cases of Uniformity of Jurisprudence. The GEMINI tool has been implemented across all twenty-four Brazilian Labour Courts, expediting the resolution of judicial cases in Brazilian Labour Court Justice [43].

The implementation of the tool in various Regional Labour Courts in Brazil has resulted in a significant reduction in the number of outstanding judicial cases. This supports the findings of this research, which indicate that grouping legal cases reduces the analysis time for assessors, thereby optimising workflow and ensuring that decisions uphold the principle of legal certainty [44–47]. Moreover, the methodology implemented through the GEMINI tool enabled faster analysis of legal proceedings by optimising the process of grouping documents, even in the absence of labels. This facilitated the work of legal advisors in a scenario of increasing demand within the Brazilian legal system, ensuring both the principle of legal certainty and the rapid processing of judicial proceedings. Furthermore, this approach contributes to advancing several Sustainable Development Goals (SDGs) defined by the United Nations, including Goal 8: "Decent Work and Economic Growth" and Goal 16: "Peace, Justice and Strong Institutions" specifically targets 8.3, 8.8, 16.3, and 16.6.

## Conclusions and future works

AI techniques for pattern detection in legal documents have proven effective, particularly when experts face large workloads. In this way, it was possible to develop, test and deploy this methodology based on deep learning for grouping judicial processes, applying it, as a case study, for the Brazilian Labour Court, enabling the use of this methodology in other languages for the legal sector, and potentially for other areas of research, as well.

Results showed this methodology to be very promising, due to the noticeable improvement in the Average Similarity Rate in the groups formed from the use of all NLP techniques applied in this work for clustering legal documents through unsupervised machine learning. In addition, this methodology dealt with documents composed of a lot of contents, which brought a difference to what has been seen in the scientific literature so far. Of all the techniques evaluated, the Sabiá 7B Jud. stands out as the best option for the generation of vector representations of documents based on the embeddings for the task of clustering legal documents of the Filed Ordinary Appeal type, however the RoBERTa Jud. technique achieved a very good result with much shorter processing time than Sabiá 7B Jud. It is important to highlight, therefore, that overall, selecting the best NLP techniques for word embeddings ensured a considerable improvement in the document groupings.

It is also worth highlighting that the techniques with fewer parameters, such as BERT, GPT-2, and RoBERTa, showed similar performance, while specialised versions (BERT Jud., GPT-2 Jud., and RoBERTa Jud.) outperformed their generalist counterparts (BERT ptBR, GPT-2 ptBR, and RoBERT ptBR). This confirms the hypothesis that model specialisation enhances results even when the corpus used in specialisation is significantly smaller than the corpus used in training the foundation model.

On the other hand, the generalist LLM model, Sabiá 7B, presented the worst result, likely due to the lack of tokenizer specialisation before fine-tuning the LlaMA model with Portuguese data. In contrast, the specialised LLM model, Sabiá 7B Jud., demonstrated superior performance, highlighting the significance of vocabulary fine-tuning and the benefits of having a higher number of training parameters.

In this way, this research presents a methodology employed in a real case involving substantial documentary content that can be adapted for various applications. It conducts a comparative analysis of existing techniques focused on a non-English language to quantitatively explain the results obtained with various NLP transformers-based models.

Therefore, future work should focus on fine-tuning a large language model (LLM) in Portuguese, such as LLaMA, by specialising its vocabulary before training, or alternatively, training a generalist LLM model in Portuguese from scratch to compare their performance in real-case scenarios. Additionally, opportunities exist to validate the word embeddings generated in this study for other types of legal documents and domains, as well as to apply them in various tasks, such as generating decision drafts and classifying documents and legal cases.

Exploring faster techniques for transforming text into vector representations is also recommended to enhance efficiency. Furthermore, the proposed methodology provides flexibility, allowing researchers to explore and apply alternative techniques, making it straightforward to substitute one approach for another. This adaptability enables the scientific community to build on this foundation, offering new perspectives and exploring alternative methods.

## Supporting information

**S1 Table. Sabiá Jud. group name.** Most relevant TPU terms for all groups generated from Sabiá Jud. model.
(PDF)

**S1 Fig. 2D Projection of groups - BERT ptBR.** Groups of documents formed using the BERT ptBR. technique, projected in two dimensions based on the test dataset.
(TIF)

**S2 Fig. 2D Projection of groups - BERT Jud.** Groups of documents formed using the BERT Jud. technique, projected in two dimensions based on the test dataset.
(TIF)

**S3 Fig. 2D Projection of groups - GPT-2 ptBR.** Groups of documents formed using the GPT-2 ptBR. technique, projected in two dimensions based on the test dataset.
(TIF)

**S4 Fig. 2D Projection of groups - GPT-2 Jud.** Groups of documents formed using the GPT-2 Jud. technique, projected in two dimensions based on the test dataset.
(TIF)

**S5 Fig. 2D Projection of groups - RoBERTa ptBR.** Groups of documents formed using the RoBERTa ptBR. technique, projected in two dimensions based on the test dataset.
(TIF)

**S6 Fig. 2D Projection of groups - RoBERTa Jud.** Groups of documents formed using the RoBERTa Jud. technique, projected in two dimensions based on the test dataset.
(TIF)

**S7 Fig. 2D Projection of groups - Sabiá 7B ptBR.** Groups of documents formed using the Sabiá 7B ptBR. technique, projected in two dimensions based on the test dataset.
(TIF)

## Acknowledgments

The authors thank the Supercomputing Centre for Industrial Innovation (CS2i) from SENAI CIMATEC (Brazil), as well as the Surrey Institute for People-Centred AI at the University of Surrey (UK), for their scientific and technical support. We also thank the Regional Labour Court of the 5th Region and the National Council for Scientific and Technological Development (CNPq, Brazil) for their support.

## Author contributions

**Conceptualization:** Raphael Souza de Oliveira, Erick Giovani Sperandio Nascimento.

**Data curation:** Raphael Souza de Oliveira, Erick Giovani Sperandio Nascimento.

**Formal analysis:** Raphael Souza de Oliveira, Erick Giovani Sperandio Nascimento.

**Methodology:** Raphael Souza de Oliveira, Erick Giovani Sperandio Nascimento.

**Software:** Raphael Souza de Oliveira.

**Supervision:** Erick Giovani Sperandio Nascimento.

**Validation:** Raphael Souza de Oliveira, Erick Giovani Sperandio Nascimento.

**Writing – original draft:** Raphael Souza de Oliveira.

**Writing – review & editing:** Erick Giovani Sperandio Nascimento.

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
