## [Decision Letter · Decision Letter 0]

13 Dec 2024

PONE-D-24-49879Analysing similarities between legal court documents using natural language processing approaches based on TransformersPLOS ONE

Dear Dr. Sperandio Nascimento,

Thank you for submitting your manuscript to PLOS ONE. After careful consideration, we feel that it has merit but does not fully meet PLOS ONE’s publication criteria as it currently stands. Therefore, we invite you to submit a revised version of the manuscript that addresses the points raised during the review process.

We look forward to receiving your revised manuscript.

Kind regards,

Hung Thanh Bui, Ph.D

Academic Editor

PLOS ONE

**Journal Requirements:**

3. Please upload a new copy of Figure 7 as the detail is not clear. Please follow the link for more information: ">https://blogs.plos.org/plos/2019/06/looking-good-tips-for-creating-your-plos-figures-graphics/"
">https://blogs.plos.org/plos/2019/06/looking-good-tips-for-creating-your-plos-figures-graphics/"

**Additional Editor Comments:**

This paper clustered 210,000 legal documents from the Brazilian justice system. The authors used k-means clustering on the document embeddings using transformer-based models: BERT, RoBERTa, GPT-2, and Llama. They found that the average within-cluster document similarity increases when they used models fine-tuned to the legal domain.

The paper lacks novelty and its experiments are not strong.

The authors should use more clustering techniques to do their experiments.

They should analyze the clusters in detail.

Comparing with another research should be done on their experiments.

Reviewers' comments:

Reviewer's Responses to Questions

**Comments to the Author**

1. Is the manuscript technically sound, and do the data support the conclusions?

Reviewer #1: Yes

Reviewer #2: Yes

Reviewer #3: Yes

Reviewer #4: Partly

2. Has the statistical analysis been performed appropriately and rigorously? 

Reviewer #1: Yes

Reviewer #2: Yes

Reviewer #3: Yes

Reviewer #4: No

3. Have the authors made all data underlying the findings in their manuscript fully available?

Reviewer #1: Yes

Reviewer #2: Yes

Reviewer #3: Yes

Reviewer #4: Yes

4. Is the manuscript presented in an intelligible fashion and written in standard English?

Reviewer #1: Yes

Reviewer #2: Yes

Reviewer #3: Yes

Reviewer #4: Yes

5. Review Comments to the Author

**Reviewer #1:** This study offers an interesting and valuable contribution to the field of NLP. Overall, the writing quality is strong and clear. However, my primary concern is that the practical application of this work within the legal sector is not clearly articulated.

While the paper provides a thorough quantitative analysis of model performance, it lacks sufficient qualitative interpretation or insights into what these results truly signify in the context of legal document analysis. Given the nature of legal work, which is inherently qualitative, it would be beneficial to include a section that explores the practical relevance of the findings from a qualitative perspective.

**Reviewer #2:** 1 Good and useful idea.

2 Good introduction.

3 Good presentation of literature review.

4 It is preferable to use the passive voice when writing the paper.

5 Figures lack high-resolution (not clear).

6 Do you think that the dataset was sufficient to produce these results? Does reducing or increasing the number of words in the vector affect the results?

7 It is preferable not to use references in the methodology or to reduce them. I think the methodology must be rewrite to be more understandable and readable.

8 The references must be written in suitable style.

**Reviewer #3: **The work "Analyzing similarities between legal court documents using natural language processing approaches based on Transformers" clusters 210,000 legal documents from the Brazilian justice system. The authors use k-means clustering on the document embeddings which they obtained by encoding the document using transformer-based models: BERT, RoBERTa, GPT-2, and Llama. The authors find that the average within-cluster document similarity increases when they use models fine-tuned to the legal domain.

My biggest concern with this work lies in the novelty of their contribution. NLP research has already proven that fine-tuning pre-trained models on the domain of interest will improve performance on downstream tasks. Therefore, we expect the "Jud"-version or specialized models to produce better semantic representations, and hence create more coherent clusters. That is not a novel finding.

Additionally, we also expect the contextualized embeddings to perform better than word2vec models because they are context-dependent. Therefore, the greater inter-cluster similarity scores in tables 3 and 4 over those in table 2 is not surprising.

Therefore, the only contribution of this paper is the application of transformer-based encoders to legal documents in the Brazilian Portuguese language. As the authors have pointed out, grouping the legal documents can assist the legal advisors in their work, as shown in Table 1. In that sense, this is more a systems paper than one doing novel methodological research. This is perfectly fine, and we do need more such works that apply conventional approaches to specialized domains and resource-constrained languages: legal Brazilian-Portuguese documents.

I suggest the authors clearly specify the contribution of this work. I recommend spending more time analyzing the clusters to find more interesting insights. For example, the authors say that Sabia produced better-defined clusters. It would be very interesting to see some examples of dissimilar documents that were clustered together by other models, but which Sabia correctly placed in different clusters, and vice-versa. Also, if the authors have some ground truth data, for example, information about legal proceedings that belong to the same group (maybe because they were deemed similar by legal advisors), they can perform an extrinsic evaluation of cluster purity.

I would also suggest that the authors should compare the clustering performance of the models on the same number of clusters (k). For any clustering approach, the average inter-cluster similarity improves as the number of clusters increase. Therefore, it is unfair to compare the similarity scores between models if they cluster the documents into different number of clusters.

I also suggest that the authors provide more details of the fine-tuning process. They should tell us which objective functions they used. We would expect masked token training to be one of the objective functions. Did they also use next-sentence prediction? Fine-tuning encoder architectures such as BERT and RoBERTa is different from encoder-decoder architectures such as GPT-2 and Llama.

**Reviewer #4: **In this paper, the authors developed a legal document clustering pipeline, in order to support efficient case distributing and reviewing. Transformer-based language models were trained to generate domain-specific contextual word embedding and compute document embedding, which were further used for the k-means clustering of legal documents. The authors implemented the proposed pipeline on a large, real-world law case data set, and compared their performance with traditional NLP and embedding methods. The background was stated in detail. As the traditional manual review for law cases requires extensive time and effort, AI-based models have good potential in real-world application. Both encoder-only and decoder-only transformer architectures were considered in this study, which provides a comprehensive comparison. Despite the strength mentioned above, I have the following questions / concerns.

1) Document grouping mechanism. In the Introduction section, the authors mentioned that grouping cases with similar requests can improve the efficiency. Why was clustering chosen rather than classification? By labeling typical types of such requests as different case classes, deep document classifiers can be fine-tuned in a supervised manner to gain more accurate and interpretable predictions of classes (groups). Different levels of classes can be designed to train multiple classifiers as well. By having the class information, certain types of cases can be assigned to those team members with specific domain expertise. If clustering is used, extra effort will be needed to interpret each cluster of cases. The authors need to better explain the motivation and real-world advantages of using clustering rather than classification in this application domain.

2) Cluster interpretation results and discussions. One key step in a clustering pipeline is to interpret the meaning of clusters, which is missing in this study. The authors chose from 30 to 61 as the number of clusters, does each cluster have practical meaning (e.g., a specific type of request, as mentioned in the Introduction)? Related to 1), clustering results are not naturally interpretable as classification, and it is very important to discuss the semantic meaning of each cluster, in addition to reporting the similarity.

3) Data set. The paragraph of Line 98 introduces the large data set used, and it is important to illustrate the statistics of the data in detail. For example, roughly how many types of legal requests or cases are there in this data set? Do these types align with the cluster number (30-61) the authors selected? Document length is important in this study as the authors carefully designed the segmenting methods. What is the average document token number in this data set? A histogram of documents with respect to token number can help readers understand the characteristics of data. Such key statistics above are missing in the manuscript.

4) Language model. Llama 3.1, 3.2 and 3.3 series all have pre-trained LLMs with 128K context windows. Specifically, Llama 3.1 has 8B (https://huggingface.co/meta-llama/Llama-3.1-8B) version and 3.2 has 1B/3B versions which are of the same size level as the models authors used in this study. These new models are multilingual and natively support Portuguese. Related to 3), if the majority of those legal documents have less than 128K tokens, do we still need to segment the document as proposed in the paper, when we choose these latest models or future models with even larger context windows?

5) Clustering method. The authors selected k-means as the clustering method. While k-means is commonly used, other clustering methods may have extra advantages in this application. Hierarchical clustering can group the cases in different levels of clusters, which naturally fits the layered relationships and nested categorizations in the legal domain. Spatial clustering methods such as DBSCAN can naturally filter out and identify noise or unusual law cases, which can be helpful in large and varied legal data. HDBSCAN incorporates both advantages above. Related to 1), the authors need to add more clarification about why k-means is chosen and how to interpret the clusters, given that there are more clustering methods with extra benefits in this application scenario.

6) In Figure 7 result, the Sabia 7B Jud. curve is a bit strange, as one specific cluster number is significantly better than all other numbers. Why is this number special? It would be valuable to interpret this specific case.

7) Writing. Things like the Python version (Line 192) should go to Experiment or Result section, as they are implementation details rather than Methodology. In Line 383, “Using” should be in lowercase.

6. PLOS authors have the option to publish the peer review history of their article (what does this mean?). If published, this will include your full peer review and any attached files.

Reviewer #1: **Yes: **Kai Ding

Reviewer #2: No

Reviewer #3: **Yes: **Sabyasachee Baruah

Reviewer #4: No

---

## [Author Response · Author response to Decision Letter 1]

27 Jan 2025

We would like to thank the editor and reviewers of this study who provided valuable suggestions and very pertinent comments, thus helping to improve the quality of our paper. All suggestions were considered and are available in the file "Response to Reviewers.pdf" submitted with this submission.

---

## [Decision Letter · Decision Letter 1]

16 Feb 2025

Analysing similarities between legal court documents using natural language processing approaches based on Transformers

PONE-D-24-49879R1

Dear Dr. Sperandio Nascimento,

We’re pleased to inform you that your manuscript has been judged scientifically suitable for publication and will be formally accepted for publication once it meets all outstanding technical requirements.

Kind regards,

Hung Thanh Bui, Ph.D

Academic Editor

PLOS ONE

Additional Editor Comments (optional):

The authors have revised the paper by my comments and all reviewers.

I accept the revision.

Please check all formatting, tables, figures and formulas.

Reviewers' comments:

Reviewer's Responses to Questions

**Comments to the Author**

1. If the authors have adequately addressed your comments raised in a previous round of review and you feel that this manuscript is now acceptable for publication, you may indicate that here to bypass the “Comments to the Author” section, enter your conflict of interest statement in the “Confidential to Editor” section, and submit your "Accept" recommendation.

Reviewer #1: All comments have been addressed

Reviewer #2: All comments have been addressed

Reviewer #4: All comments have been addressed

2. Is the manuscript technically sound, and do the data support the conclusions?

Reviewer #1: Yes

Reviewer #2: Yes

Reviewer #4: Yes

3. Has the statistical analysis been performed appropriately and rigorously? 

Reviewer #1: Yes

Reviewer #2: Yes

Reviewer #4: Yes

4. Have the authors made all data underlying the findings in their manuscript fully available?

Reviewer #1: Yes

Reviewer #2: Yes

Reviewer #4: Yes

5. Is the manuscript presented in an intelligible fashion and written in standard English?

Reviewer #1: Yes

Reviewer #2: Yes

Reviewer #4: Yes

6. Review Comments to the Author

Reviewer #1: The authors have made significant revisions in response to the reviewers' comments, and I find their revisions satisfactory. I have no further comments at this time.

Reviewer #2: (No Response)

Reviewer #4: The authors’ revision and response have addressed my previous questions and concerns. The challenge of legal court data annotation has been discussed in the revised version, providing a reasonable motivation for using clustering. Tables 8 and S1 have been added to interpret the meaning of clusters through the most relevant terms, and the statistics of the document length have been reported. While later language models and more sophisticated clustering methods are not available due to the computational capacity constraints, the authors have discussed the extensibility of their proposed method.

As a minor suggestion, the authors may consider adding English translations for the terms in Table S1. Since this is an English manuscript, such translations may help readers interested in the general method but unable to read Portuguese, allowing them to understand the clusters without a translator. As the table is already in the Supplement, a lengthy translation alongside the original Portuguese terms shouldn’t be an issue.

No further comments from my side.

7. PLOS authors have the option to publish the peer review history of their article (what does this mean?). If published, this will include your full peer review and any attached files.

Reviewer #1: **Yes: **Kai Ding

Reviewer #2: No

Reviewer #4: No

---

## [Editor Report · Acceptance letter]

PONE-D-24-49879R1

PLOS ONE

Dear Dr. Sperandio Nascimento,

I'm pleased to inform you that your manuscript has been deemed suitable for publication in PLOS ONE. Congratulations! Your manuscript is now being handed over to our production team.

Kind regards,

on behalf of

Dr. Hung Thanh Bui

Academic Editor

PLOS ONE